# Structural Analysis of the *Drosophila melanogaster* GSTome

**DOI:** 10.3390/biom14070759

**Published:** 2024-06-26

**Authors:** Nicolas Petiot, Mathieu Schwartz, Patrice Delarue, Patrick Senet, Fabrice Neiers, Adrien Nicolaï

**Affiliations:** 1Laboratoire Interdisciplinaire Carnot de Bourgogne, UMR 6303 CNRS–Université de Bourgogne, 21078 Dijon, France; nicolas.petiot01@u-bourgogne.fr (N.P.); delaruep@u-bourgogne.fr (P.D.); psenet@u-bourgogne.fr (P.S.); 2Flavour Perception: Molecular Mechanims (Flavours), INRAE, CNRS–Université de Bourgogne, 21000 Dijon, France; mathieu.schwartz@inrae.fr (M.S.); fabrice.neiers@u-bourgogne.fr (F.N.)

**Keywords:** glutathione transferase, *Drosophila melanogaster*, AlphaFold, multiple sequence alignment, dimerization interface, binding sites, normal mode analysis

## Abstract

Glutathione transferase (GST) is a superfamily of ubiquitous enzymes, multigenic in numerous organisms and which generally present homodimeric structures. GSTs are involved in numerous biological functions such as chemical detoxification as well as chemoperception in mammals and insects. GSTs catalyze the conjugation of their cofactor, reduced glutathione (GSH), to xenobiotic electrophilic centers. To achieve this catalytic function, GSTs are comprised of a ligand binding site and a GSH binding site per subunit, which is very specific and highly conserved; the hydrophobic substrate binding site enables the binding of diverse substrates. In this work, we focus our interest in a model organism, the fruit fly *Drosophila melanogaster* (*D. mel*), which comprises 42 GST sequences distributed in six classes and composing its GSTome. The goal of this study is to describe the complete structural GSTome of *D. mel* to determine how changes in the amino acid sequence modify the structural characteristics of GST, particularly in the GSH binding sites and in the dimerization interface. First, we predicted the 3D atomic structures of each GST using the AlphaFold (AF) program and compared them with X-ray crystallography structures, when they exist. We also characterized and compared their global and local folds. Second, we used multiple sequence alignment coupled with AF-predicted structures to characterize the relationship between the conservation of amino acids in the sequence and their structural features. Finally, we applied normal mode analysis to estimate thermal B-factors of all GST structures of *D. mel*. Particularly, we extracted flexibility profiles of GST and identify key residues and motifs that are systematically involved in the ligand binding/dimerization processes and thus playing a crucial role in the catalytic function. This methodology will be extended to guide the in silico design of synthetic GST with new/optimal catalytic properties for detoxification applications.

## 1. Introduction

Glutathione transferase (GST) is a superfamily of enzymes that are ubiquitous and multigenic in numerous organisms. Originally discovered as detoxification enzymes [1], GSTs are one of the key proteins involved in the metabolism of endogenous and exogenous molecules with hydrophobic character. Therefore, they are involved in many biological functions, such as chemoperception [2], isomerases [3], mediators in oxidative stress responses [4], and more [5]. The vast majority of GSTs are active as homodimers and are classified in three main families depending on their location in cells: mitochondrial, microsomal, or cytosolic (canonical) [6]. The cytosolic GST family, which is the most abundant one, contains four different classes named Theta, Omega, Sigma, and Zeta. Moreover, organisms in different taxa present additional classes. For instance, mammals present GST classes named Alpha, Mu, and Pi, whereas insects present GST classes named Delta and Epsilon.

The main biological function of cytosolic GST is to catalyze the conjugation of their cofactor, the reduced glutathione (γ–Glutamyl–Cysteinyl–Glycine, named GSH) to xenobiotic electrophilic centers. To achieve this catalytic function, GSTs are comprised of a GSH ligand binding site, named the G site (GS), and a hydrophobic substrate binding site, named the H site (HS), with one G and one H site per monomeric subunit. The G site is very specific to GSH, whereas the H site allows the binding of a large variety of substrates. Moreover, residues forming the G site are particularly conserved among GST, especially Tyrosine and Serine amino acids, that act as hydrogen-bond donors for the thiol group of GSH [6] (Tyr-GSTs and Ser-GSTs). Residues involved in the H site of xenobiotic substrates are mainly comprised of non-polar side chains. Their positions in the sequence and the nature of these residues are much more substrate-dependent and do not always present a high conservation score among GST sequences of different classes, families, or organisms. Structurally speaking, the two binding sites are located in two well-separated characteristic domains of GSTs. The N-terminal domain, i.e., domain I, is a thioredoxin-like fold made of $\beta_{1}$-$\alpha_{1}$-$\beta_{2}$-$\alpha_{2}$-$\beta_{3}$-$\beta_{4}$-$\alpha_{3}$ secondary structures (Figure 1A). The second domain, i.e., domain II, corresponds to the C-terminal and is mainly formed by α helices. During evolution, cytosolic GST were obtained by domain addition to the thioredoxin fold. In all dimeric structures of GST, inter-monomer interactions occur between domain I of monomer A and domain II of monomer B (Figure 1C). The type of interactions involved in the dimerization interface (DI) is class-dependent. For example, GST classes Delta and Epsilon are characterized in most structures by a lock and key, “Clasp” [7] or “Wafer” [8] motifs, based on electrostatic interactions, whereas GST classes Theta and Sigma present interactions that are mostly hydrophilic. It has been demonstrated that, for some GSTs, a single mutation in these motifs is able to break these interactions and therefore significantly reduces the catalytic efficiency of the enzyme [9].

In the present work, the *Drosophila melanogaster* organism (*D. mel*, fruit fly) is used as a model organism, as it has been in several works [10,11,12]. Due to their relatively small size, the insects have developed multiple mechanisms to limit toxic effects, through the enhancement of metabolic detoxification [13]. Moreover, interactions between insects and plant chemicals lead to a major driving force in herbivorous insect evolution, leading to very efficient and versatile enzymes. For instance, it has been demonstrated that GSTs have a role in the development of resistance to pesticides in plants and animals [14,15]. An increase in the catalytic activity of GSTs enhances the ability of the organism to detoxify noxious chemicals. The chemical diversification in a plant during the evolution was probably an important evolutive driver leading to GST diversification, especially in insects. This encourages the study of insect GST to understand how spontaneous mutations/insertions/additions in the sequence modify the stability, selectivity, and the efficiency of this superfamily of enzymes. The Universal Protein Database [16] (UniProt) registered a total of 36 *gst* genes, encoding 42 GST proteins which compose the *D. mel* GSTome [12,17]. They are distributed in six classes, with classes Delta and Epsilon being the largest classes with 11 and 14 members, respectively; classes Omega, Sigma, Theta, and Zeta comprise 4, 1, 4, and 2 members, respectively. The six missing protein structures are isoforms, distributed in classes Delta, Theta, and Zeta. As the purpose of the present study is based on the complete structural analysis of *D. mel* GSTome, we only considered the 36 GST proteins corresponding to the 36 *GST* genes of the *D. mel* organism. These 36 proteins are all homodimeric structures and cytosolic GST.

Nowadays, bio-informatics techniques have been extensively used to study protein sequence/structure relationships and they provide valuable indicators of protein biological functions [18]. The most widely used technique is the multiple sequence alignment (MSA), which consists of aligning three or more biological sequences by addition of “gaps”. From MSA output, similarity and evolutionary relationships can be deduced. In addition to sequence based methods, other techniques such as solvent accessibility [19] or secondary structure prediction [20], which are based on analysis of protein atomic structures, can be used. Historically, protein structures were only accessible from experiments using X-ray diffraction (XRD), nuclear magnetic resonance (NMR), and cryogenic electron microscopy (cryo-EM), or from homology modeling [21]. Very recently, artificial-intelligence-based techniques such as AlphaFold [22] or RoseTTAfold [23] were developed to make three-dimensional (3D) structural predictions of proteins at the atomic level, with accuracy approaching the ones obtained from experiments. It therefore represents a gigantic step forward for the structural biology community. Among the 36 GST sequences of *D. mel* organism mentioned above, only 8 of them have been solved experimentally and deposited in the Protein Data Bank (Table 1). Therefore, the use of AI algorithms to predict 3D atomic structures of the complete *D. mel* GSTome is needed. Only one study reports the analysis of a complete GSTome, based on phylogenic analysis, expression, purification, and substrate specificity but not from atomic structures [11]. Recently, a similar approach was applied to characterize the human GSTome [24].

In the present work, we performed the prediction of the 36 homodimeric 3D atomic structures of *D. mel* using AlphaFold-multimer program [25]. Additionally, we used the AlphaFill program [26] to transplant GSH ligands into the predicted structures. Moreover, we applied multiple sequence alignment coupled with AlphaFold predictions to characterize three main structural features of GST enzymes, i.e., their global and local folds using secondary structure predictions, the composition of their GSH binding sites, and the dimerization interface. We identified residues that are systematically involved in the ligand binding and dimer interactions, which play a crucial role in the catalytic function, and discuss their conservation in the *D. mel* GSTome. Finally, we performed normal mode calculations using the anisotropic network model [27] to predict thermal B-factors of all GSTs from *D. mel*. From this analysis, we extracted flexibility profiles of the complete GSTome and for each class of GST. This methodology will be extended to guide the in silico design of synthetic GST enzymes with new/optimal catalytic properties for depollution applications.

**Table 1 biomolecules-14-00759-t001:** Summary of GST structures resolved experimentally using XRD and accessible from the Protein Data Bank.

Name	UniProt	Sequence Length	PDB	Number of Monomer	Monomer Length	Resolution [Å]
GSTD1	P20432	209	3ein [4]	1	195	1.13
			3mak [28]		208	1.80
GSTD2	Q9VG98	215	5f0g [29]	2	197	1.60
					204	
GSTD10	Q9VGA1	210	3f6f [28]	1	209	1.60
			3gh6 [28]		208	1.65
GSTE1	Q7KK90	224	9f7k [30]	2	219	1.74
					217	
GSTE6	A1ZB71	222	4yh2 [8]	4	219	1.72
					220	
					221	
					221	
			4pnf [31]	8	220	2.11
					221	
					219	
					219	
					220	
					220	
					219	
					219	
GSTE7	A1ZB72	223	4png [31]	2	221	1.53
					223	
GSTE14	Q7JYX0	232	6kel [32]	2	215	1.40
					221	
			6kem [32]	2	200	1.50
					208	
			6t2t [33]	1	229	1.30
GSTS1	P41043	249	1m0u [34]	2	203	1.75

## 2. Materials and Methods

### 2.1. Multiple Sequence Alignment of Drosophila melanogaster GSTome

Sequences of *D. mel* GSTome were extracted from the UniProt database and multiple sequence alignment using the 36 corresponding sequences was performed using the Clustal Omega program [35]. A residue in the MSA array is identified by a GST sequence (i.e., row-wise) and a number (i.e., column-wise). A residue at a given MSA number is called conserved if the exact same amino acid is present at the exact same number in the 36 sequences constituting the GSTome. In addition, sequence identity is defined as a pairwise comparison between GST sequences. It is given in percent and is computed from pairwise residue conservation in two aligned sequences. Basically, an identity of 100% corresponds to two sequences that are exactly identical and an identity of 0% corresponds to two sequences with no identical residues at the same position in their sequences. When compared to the alignment score that uses substitution matrices, sequence identity only identifies the exact match between pairs of residues.

### 2.2. Prediction of 3D Atomic Structures of Drosophila melanogaster GSTome

Three-dimensional atomic structure predictions were performed from the ColabFold project [36], which uses the AlphaFold-multimer-v3 program [25]. From the FASTA sequence of each GST, the generated output comprises five predicted structure files, including 3D Cartesian coordinates of all heavy atoms (N, C, O, S) of each amino acid along the sequence. The five corresponding structures are sorted in decreasing “prediction score”, the so-called predicted local-distance difference test [37] (pLDDT). It is a per-residue measure of local confidence. It is scaled from 0 to 100, with higher scores indicating higher confidence and usually a more accurate prediction. When compared to the root mean squared deviation of atomic positions (RMSD), pLDDT is not dependent on structural alignment and allows pairwise comparisons, even in the presence of domain motions. In this work, we only consider the structure with the highest pLDDT score for each GST as the predicted structure. Next, we used the AlphaFill program [26] to predict the position of GSH ligands in the binding site (G site). AlphaFill uses sequence and structural similarities between proteins to transplant ligands, from experimental to theoretical structures predicted by AlphaFold. It searches for sequence homologs that have a sequence identity above 25% in its database and performs structural alignment. The corresponding transplanted ligands are sorted by decreasing sequence identity between the query and the database. As an example, for GSTD11iA, AlphaFill predicted 1 GTS ligand (Glutathione Sulfonic Acid) with sequence identities larger than 50%. To consider a larger ensemble of ligand positions, we took into account ligands extracted from GST experimental structures with a sequence identity larger than 40%. It leads to a total of 9 GSH-like ligands per GSTD11iA monomer, with 1 GSF and 8 GTX (S-Hexylglutathione). Then, we performed a clustering based on distances between the different ligands to extract relevant binding positions, with a distance threshold of 2.0 Å between ligands in a given binding site. For GSTD11iA dimeric structure as described above, it corresponds to only two different binding sites, each of them containing nine ligands. The complete details about AlphaFill predictions for the other GST structures of *D. mel* are given in the Appendix A.

### 2.3. Characterization of Global and Local Conformations of GST Structures from Drosophila melanogaster and Comparison with Available Experimental Data

From 3D atomic structures predicted by AlphaFold, analysis of pLDDT score as a function of MSA number and prediction of secondary structures using the CUTABI algorithm [38] were performed. It allows us to determine which parts of the structures are well (or badly) predicted by AF and which secondary structures are associated with it. Moreover, to gain insights into the structural similarities/differences existing in the *D. mel* GSTome, we defined global and local metrics to characterize conformational changes. First, we computed the global deviation between pairs of dimeric GST structures as the RMSD of atomic positions. Technically, RMSD was computed using the positions of $C^{\alpha }$ atoms (3D Cartesian coordinates), which are common to the two sequences of interest (no gap), after the alignment of the two structures using PyMOL [39]. Second, we computed local deviation along the amino acid sequence between pairs of GST structures as the distance between the positions of $C^{\alpha }$ (3D Cartesian coordinates). It corresponds to a local information compared to the global deviation, which provides information about 3D global fold changes. Finally, local deviation was also computed using internal coordinates, i.e., angles, which are therefore independent of structural alignment. As we performed in a previous work [40], coarse-grained angles (CGAs) θ and γ were used to characterize the local conformation of protein main chain. Assuming a constant virtual bond length between $C^{\alpha }$ atoms of successive residues, a chain of *N* amino acids is fully characterized by N−3 CGA: torsion angles $\gamma_{n}$ built from $C_{n-1}^{\alpha }$, $C_{n}^{\alpha }$, $C_{n+1}^{\alpha }$, and $C_{n+2}^{\alpha }$; and bond angle $\theta_{n}$ built from $C_{n-1}^{\alpha }$, $C_{n}^{\alpha }$, and $C_{n+1}^{\alpha }$, with n=2 to N−2. These CGA are, respectively, the discrete version of the local curvature ($\theta_{n}$) and the local torsion ($\gamma_{n}$) of the main chain formed by successive $C^{\alpha }$–$C^{\alpha }$ virtual bonds [41]; they are used to define secondary structures based only on $C^{\alpha }$ coordinates [38]. Dihedral angles $\chi_{1}$ were used to characterize the local conformations of the side chains.

### 2.4. Characterization of the Dimerization Interface and of the GSH Binding Sites of Drosophila melanogaster GSTome

Contact maps of GST structures including their GSH cofactors were computed using 3D Cartesian coordinates of all heavy atoms. The goal was to identify which amino acids belong to: (i) the dimerization interface (DI), i.e., residues of a GST dimeric structure that are in close contact between the two monomeric subunits; (ii) the GSH binding sites (GSs), i.e., the residues of a GST enzyme that are in contact with GSH cofactors predicted by AlphaFill. Technically, two atoms were considered in contact if they were separated by a distance smaller than 4.0 Å. Then, two residues, *i* and *j*, were considered in contact if at least one atom of residue *i* was in contact with one atom of residue *j*. In addition, we used Dijkstra’s algorithm [42] to map communication pathways between the two GSH binding sites (from the G site of monomer A to the G site of monomer B). From this analysis, we identified atoms of GST that are in contact with atoms of GSH ligand and then we computed all the possible pathways between the two binding sites. This analysis was performed in an asymmetrical configuration, i.e., with a GSH-bound binding site in monomer A and an empty binding site in monomer B. Finally, energies (in kJ/mol) and surfaces (in Å^2^) of the dimerization interface of each GST were computed using molecular dynamics. All-atom MD calculations were carried out with the GROMACS software package [43] (version 2018.2, double precision), using the CHARMM36 force-field [44,45]. A single point calculation was performed to extract the potential energy of the two individual monomeric subunits and of the dimeric structure, after an energy minimization in vacuum of the 3D atomic structures predicted by AlphaFold. The steepest descent algorithm was used with a maximum step size of 0.01 nm and a tolerance of 1000 kJ/mol/nm.

### 2.5. Structural Dynamics Predictions of Drosophila melanogaster GSTome from Normal Mode Calculations

Normal mode analysis was performed using the anisotropic network model (ANM), which is a simple yet powerful model to predict the structural dynamics of proteins [27,46]. In this model, 3D atomic structures of proteins are represented as an elastic network of masses-and-springs. In the corresponding network, each node is a heavy atom of the protein (N, C, O, S) and springs represent the physical interaction between the nodes. In ANM, the mass-weighted Hessian matrix $\hat{H}$ for a network of *N* nodes is a 3N×3N matrix of the following form:(1)$${\hat{H}}_{ij}^{\alpha \beta }=\frac{H_{ij}^{\alpha \beta }}{\sqrt{m_{i}m_{j}}}\;\mathrm{with}\;H_{ij}^{\alpha \beta }=-\frac{R_{ij}^{\alpha }R_{ij}^{\beta }}{R_{ij}^{2}}\Gamma_{ij}\;;\;H_{ii}=-\sum_{j\neq i}H_{ij}$$
where α and β represent one of the Cartesian directions x,y, or *z*; $R_{ij}$ is the distance between two nodes *i* and *j*; $m_{i}$ is the mass of the node *i*; $\Gamma_{ij}=\gamma /R_{ij}^{2}$ with $\gamma =\partial^{2}V/\partial R_{ij}^{2}$ is the force constant of the elastic bond between the nodes, i.e., the so-called parameter-free ANM [47]. Normal modes are then obtained by performing the diagonalization of the mass-weighted Hessian matrix:(2)$$\hat{H}e_{k}={\tilde{\omega }}_{k}^{2}e_{k}$$
with 3N−6 non-zero eigenfrequencies ${\tilde{\omega }}_{k}$ and their corresponding eigenvectors $e_{k}$. From the eigenfrequencies and eigenvectors of the mass-weighted Hessian matrix, we computed thermal B-factors [48], as measured experimentally using XRD, and corresponding to:(3)$$B_{i}=\frac{8\pi^{2}}{3}\frac{k_{B}T}{m_{i}}\sum_{k=1}^{3N-6}\frac{|e_{ki}{|}^{2}}{{\tilde{\omega }}_{k}^{2}}$$

## 3. Results and Discussion

### 3.1. Comparison of GST Sequences from Drosophila melanogaster Using Multiple Sequence Alignment

Figure 2A presents a sample of multiple sequence alignment generated using the 36 GST sequences from *D. mel*; the full MSA is given in the Appendix A. First, in the GSTome, sequence lengths vary between 199 (GSTD3) and 268 residues (GSTT3); this means that there is potentially a large number of gaps between GST sequences. As shown in Figure 2B, the number of gaps can reach the value of 35. By looking at the probability distribution, two distinct peaks were observed, one for a very few number of gaps (between 0 and 5 gaps) and one for a very large number of gaps (between 28 and 35 gaps). Overall, around 45% of GST sequences do not present a gap and 63% present 6 gaps or less. Per class, sequence lengths vary from 199 to 224 residues for class Delta; from 220 to 240 for class Epsilon; from 241 to 254 for class Omega; from 228 to 268 for class Theta; from 227 to 246 for class Zeta. GSTS1 enzyme is the only GST of class Sigma and has a sequence length of 249 residues. GSTs, which belong to class Delta, are shorter than the others, inducing several gaps when comparing two sequences from a different class. This is especially true for MSA numbers <50, associated with the N-terminal (Figure 2B). MSA numbers 118–121 and 249–256 also present a large number of gaps between GST sequences due to specific insertions in the evolution of class Omega. By analyzing the complete MSA, only three MSA numbers are associated with a perfect 100% conservation score. They correspond to Pro113, which have been found to be an essential constituent of the G site of GSTs [6,32]; Leu136 and Asp237, which is known to be part of a N-capping box motif [49]. These last two residues are located in $\alpha_{3}$ and $\alpha_{6}$ helices of the structure. For all the other MSA numbers, the conservation score is less than 100% and can vary from 14% (MSA number 63) to 97% (MSA number 132). Similarlly, the total charge of GST proteins in the *D. mel* GSTome can strongly vary, from −15 (GSTS1) to +17 (GSTZ1). In GSTS1, 11 negatively charged amino acids are located in the N-terminal part (MSA number < 50); this region includes a large number of gaps in the MSA, as shown in Figure 2B. In GSTZ1, six positively charged amino acids are located in the N-terminal, and most of the others are located in domain I which has a total charge of +12 (MSA number < 140). On average, GST classes Delta and Sigma are globally negatively charged, whereas classes Epsilon, Omega, Theta, and Zeta are globally positively charged. However, the total charge of GST in class Epsilon is on average null and GST class Zeta is the most positively charged class among the six of *D. mel.* GSTome. Moreover, for class Delta, with the exception of GSTD8 (−11), the total charge is between −4 (GSTD6) and +2 (GSTD1); for class Epsilon, the total charge is between −8 (GSTE4) and +6 (GSTE10); for class Omega, the total charge is between +3 (GSTO3) and +7 (GSTO1 and GSTO2iA); for class Theta, the total charge is between +6 (GSTT3) and +10 (GSTT4); for class Zeta, the total charge is between +8 (GSTZ2) and +17 (GSTZ1). The exact location of positively and negatively charged amino acids as a function of MSA number for the complete GSTome is given in the Appendix A.

Figure 2D presents four examples of conservation scores for two well-known biological features of GST. First, the active residue of the G site is characterized by a Cystein for all four sequences in class Omega and by a Serine for most of the other classes. This residue has been identified in domain I (N-terminal) at MSA number 64. The conservation plot shows a high score for Serine (≈72%). Exceptions concern Gly64 for GSTD2, which presents the unique Glycine patch among GST [29] and for GSTD5, which does not correspond to a Glycine patch. GSTD7 is characterized by an Alanine, GSTS1 by a Lysine, and GSTT4 by an Aspargine at MSA number 64. Second, a central lock and key motif of GST has been identified for enzymes in classes Delta and Epsilon. For GST class Delta, this key motif, referred as the “Clasp” motif, comprises Tyr165 and Met168 [28]. For GST in class Epsilon, this motif is different and is referred as the “Wafer” motif (Figure 1). It comprises His130, His165, and Ser168 [8]. For these three MSA numbers, the conservation score is below 40%, with Arginine and Histidine residues at MSA number 130; but, interestingly, we found Leucine residues that are specific to class Omega, and Valine and Isoleucine residues that are specific to the other classes of *D. mel*. The same observation is performed for Tyr165, which is specific to class Delta (except for GSTD11iA with a Glutamine). Class Epsilon is mostly characterized by Histidine residues at this MSA number, except for GSTE1 and GSTE2, characterized by a Phenylalanine of motif “Clasp+Wafer” [30]; GSTE11 and GSTE14 being characterized by an Histidine and an Isoleucine, respectively. GST classes Omega, Sigma, Theta, and Zeta are mostly characterized by negatively charged amino acids at MSA number 165, i.e., Aspartic acid, except for GSTT4 (Alanine). MSA number 168 is the less conserved among motif residues with a large variability in the nature of the amino acid observed for the 36 sequences of *D. mel*, with Methionine and Histidine residues being the most conserved for classes Delta and Epsilon, respectively. Interestingly, Cystein residues are also identified for class Zeta, which are probably not directly related to the binding of GSH. However, it may have a crucial role in the stabilization of GST dimeric structure for this class specifically (see Section 2.4). Finally, we compared sequence identity between pairs of GST sequences. As shown in Figure 2E, sequence identities vary from 11% for the pair GSTS1–GSTT4 to 79% for the pair GSTD2–GSTD5. Moreover, sequence identities within a class vary between 45% and 64%. GST classes Delta and Epsilon show very similar sequences when compared to each other, in contrast to the other classes, with 34% sequence identity on average and below 20% for the other classes. Clusters of sequence identity within a class were identified: GSTD1 to GSTD5 for class Delta and GSTE5 to GSTE8 for class Epsilon. GSTD11iA, GSTE14, and GSTT4 present sequence identities with the other classes that are much lower than the other members.

### 3.2. Analysis of the Global and Local Conformations of GST from Drosophila melanogaster and Comparison between AlphaFold Predictions and Experiments

From the 36 sequences identified in *D. mel* GSTome, only 12 structures corresponding to 8 different GSTs have been resolved experimentally using X-ray crystallography and deposited in the Protein Data Bank (Table 1). In detail, *3ein* [4], *3mak* [28] (GSTD1), *3f6f, 3gh6* [28] (GSTD10) and *6t2t* [33] (GSTE14) PDB structures contain one monomer per asymmetric unit, whereas *5f0g* [29] (GSTD2), *4png* [31] (GSTE7), *6kel, 6kem* [32] (GSTE14), and *1m0u* [34] (GSTS1) PDB structures contain dimeric structures. In addition, *4yh2* [8] and *4pnf* [31] (GSTE6) are PDB complexes of two and four dimeric structures, respectively. GST experimental structures are given in the Appendix A. Except for *1m0u* (GSTS1), XRD structures present a different number of residues in their monomeric subunits. When compared to their gene sequence length given by UniProt, the number of residues missing in XRD structures can vary from 1 (*3mak*, GSTD1 and *3f6f*, GSTD10) to 46 amino acids (*1m0u*, GSTS1). Particularly, 32 and 46 residues are missing for *6kem* (GSTE14) and *1m0u* (GSTS1), respectively. Missing parts are located in the N-terminal, which have been either removed before crystallization or had too low electron density to be detected. Historically, in 2003, Agianian et al. published GSTS1 structure (*1m0u*), proposing that GST class Sigma is not related to the detoxification but rather to protect tissues from oxidative stress [34]. It has also been shown that GSTS1 from *D. mel* has low activity toward the commonly used synthetic substrate 1-chloro-2,4-dinitrobenzene (CDNB), but has relatively high glutathione-conjugating activity for 4-hydroxynonenal (4-HNE), an electrophilic aldehyde derived from lipid peroxidation [50]. In 2010, Low et al. studied recognition and detoxification of the insecticide DDT by GSTD1. They observed that $\alpha_{8}$ helix occluded the H site in its APO form while still being able to metabolize DDT [4]. In 2012, Wongsantichon et al. observed structural transition after ligand binding in both GSTD1 and GSTD10 [28]. In 2015, Scian et al. compared GSTE6 and GSTE7 structures, concluding that they were structurally and functionally comparable [31]. In addition, it has also been shown that GSTE6 and GSTE7 display outstanding catalytic activities with several substrates, particularly with environmental pollutants [51,52]. In 2018, Gonzalez et al. studied GSTD2 and observed a unique “GGGG” motif called the Glycine patch [29]. In 2020, Koiwai et al. studied GSTE14 and its implication in ecdysteroid synthesis [32]. In 2020, Skerlová et al. demonstrated that GSTE14 is catalytically active with steroid substrates [33]. Finally, in 2024, Schwartz et al. studied GSTE1 and characterized its dimerization interface from the XRD structure. Particularly, they showed that the interface contains a central lock and key motif that is an hybrid motif between “Clasp” and “Wafer” [30]. To summarize, all the experimental structures of GST from *D. mel* represent in total only 8 out of the 36 sequences of the complete GSTome.

Therefore, we performed structural predictions of GST proteins including their GSH cofactors for the 36 sequences of *D. mel*, as explained in Section 2.2. Figure 3A presents AlphaFold prediction scores (pLDDT) as a function of MSA number. Overall, pLDDT score along the sequence is larger than 95% for all GSTs, with two main exceptions observed for N- (MSA number <50) and C-terminal domains (MSA number >310). In addition, GSTO1 also shows prediction scores ≈60% for MSA numbers 175–210 and 240–260, associated with the loop between $\alpha_{4}$ and $\alpha_{5}$ helices and to the end of $\alpha_{6}$ helix, respectively. This structural part of GST only exist for class Omega, with gaps in the MSA for the other classes (Figure 3A). The corresponding 3D structures are shown in Figure 3B and in the Appendix A and secondary structure prediction as a function of MSA number is presented in Figure 3C. Although the 3D folds of GST look very similar, the local structural properties can vary in the *D. mel* GSTome due to additions/insertions/deletions during the evolution. First, the N-terminals of the GSTS1 and GSTT1 structures are predicted as loops that are intrinsically disordered and that have been badly predicted by AlphaFold (pLDDT ≈30%). Second, the thioredoxin-like fold, indicated by the $\beta_{1}$-$\alpha_{1}$-$\beta_{2}$-$\alpha_{2}$-$\beta_{3}$-$\beta_{4}$-$\alpha_{3}$ labels in Figure 3C, is very well conserved. Third, as mentioned in Section 3.1, GSTD3 is characterized by the shortest sequence among all GSTs of *D. mel* and is associated with a missing β-sheet and a cropped α helix in its N-terminal (domain I) compared to the other structures, starting at MSA number 72. Consequently, GSTD3 is missing Ser64, which is characteristic of the active site in GST. In addition, GSTD7 is characterized by a longer $\alpha_{8}$ helix when compared to other GSTs and classes Omega and Zeta exhibit structures with shorter $\alpha_{2}$ helix. Aside from that, all GSTs are characterized by identical secondary structures in domain I. The linker loop between domains I and II is located at MSA numbers 140–153. Domain II is the one that shows the biggest differences in secondary structure predictions compared to domain I, which is more conserved through evolution. For example, we identified potentially five α helices that have a relatively well defined positions in the sequence but with some variations between GSTs. In addition, part of the structure between MSA numbers 168 and 176 contains loops that shorten the $\alpha_{4}$ helix for GSTE4–GSTE11. This is the range of MSA numbers that corresponds to the “Wafer” motif in the dimerization interface, as mentioned above. From 3D structures, we observed that these interactions between monomeric subunits in this region seems to deform the α helix, leading to a partial local unfolding. In addition, this structural feature is also visible in the structures of GST in classes Delta, Theta, and Zeta but for a fewer number of residues. Moreover, C-terminal part of GSTE10 (MSA numbers 310–332) presents longer $\alpha_{8}$ helix compared to the other GSTs, which is oriented towards the outside of the structure (Figure 3B). However, this local structure has a pLDDT score that is much lower than for the other regions of the protein (<60%). Finally, a set of two extra β sheets located between $\alpha_{4}$ and $\alpha_{5}$ helices on one side, and in the C-terminal on the other side, were found in GSTE1–GSTE9 (Figure 3C). These additional secondary structures are very specific and may have an impact on the dynamics of GST structures, introducing extra local rigidity. Identically, a short $\beta_{0}$ sheet of two residues was predicted for GST in class Omega with a very high pLDDT score from AlphaFold, whereas an extra $\alpha_{0}$ helix was also predicted in the N-terminal part of GSTT3 but with a prediction score below 50% (Figure 3B).

For all pairs of AF-predicted structures, we computed the global deviation of the backbone between GST proteins. Figure 4A shows a matrix representation of the pairwise computed RMSD. Overall, RMSD values vary between 0.8 (pair GSTE5–GSTE6) and 10.0 Å (pair GSTE10–GSTO1). Since thermal fluctuations are associated with values of the order of magnitude of ≈2.0 Å, pairs of GST structures with a pairwise RMSD lower than 2.0 Å are considered having an identical 3D fold. With this criterion, identical folds were identified in GST classes Delta (11 members) and Zeta (2 members); class Sigma containing only 1 member. On the contrary, GST classes Omega and Theta (4 members each) present the largest global deviation within a class, whereas GST class Epsilon (14 members) is the one showing the largest variability, particularly for structures GSTE10 to GSTE14 (Figure 4A). For instance, global deviation ranges between 0.8 Å (GSTE5-E6) and 5.4 Å (GSTE10-E11). Moreover, GST structures which belong to different classes are characterized by non-similar 3D folds when compared to each other, which means that the global fold is strongly class-dependent. Fold similarity between GST classes Delta and Epsilon is the largest, with RMSD values on average around 3.8 Å compared to RMSD values on average around 6.0 Å between the other classes (Figure 4B). The two classes showing the largest differences of 3D fold within each other are GST classes Epsilon and Omega, even though they are characterized by similar sequence lengths. This observation is also correlated with the sequence similarity already observed in Figure 2E. Structurally, it comes from the fact that the dimerization interface is much smaller in GST class Omega than all the other classes, which leads to much more separated monomeric subunits and from the C-terminal domain. As shown in the Appendix A, the radius of the gyration of GST structures in class Omega is around 24 Å, which is a little bit larger than the mean value observed for the other classes (22 Å). Within a class, the radius of gyration is ≈21 Å for classes Delta and Epsilon, with the exception of GSTD7, GSTE10, and GSTE14, which are above 22 Å. GST classes Theta and Zeta are characterized by a radius of gyration ≈23 Å, with the exception of GSTZ1 (29 Å). Exceptions listed here generally have longer N- and C-terminal parts. In the case of GSTD7 and GSTE10, it is a longer C-terminal α helix. In the case of GSTZ1, it is a long N-terminal loop; for GSTE14, it is both the N- and C-terminal loops that significantly increase the radius of the gyration. For experimental structures, and as mentioned in Section 3.1, the terminal parts are usually removed during the structure resolution process.

Figure 4C shows local deviation as a function of MSA number for two selected pairs of GST structures, within a class and between GST structures from two different classes. On the one hand, the local deviation for the pair GSTE5–GSTE6 is always below 2.0 Å, which shows how similar the two folds are, both at a global and at a local scale. Structural similarities between GSTD6 and GSTD7 have already been pointed out experimentally [31] and, from the present work, the structural similarities are confirmed for GSTE5-E8 (Figure 4A). For pair GSTE10–GSTO1, the local deviation can be as large as 20.0 Å, which confirms significant differences between the two folds. Particularly, the C-terminal domain (MSA numbers 280–300) is characterized by residues with large deviations (≈10.0 Å), associated with the $\alpha_{8}$ helix on both structures. Surprisingly, MSA numbers 160–210, corresponding to $\alpha_{4}$ and $\alpha_{5}$ helices, also show large local deviations. It is probably due to the fact that monomeric subunits of GSTO1 are much more separated compared to the ones of GSTE10. MSA numbers 80–100 show different secondary structures locally, a loop for GSTO1 and an α helix for GSTE10, leading to significant conformational changes. It is noteworthy that, even if global folds present significant differences, local deviations for MSA numbers 55–95 (in the conserved domain I) are very low (≈1.0 Å). We performed the same analysis of local deviation along the amino acid sequence by comparing AF-predicted structure of GSTE7 with its corresponding XRD structure (*4png*), as shown in Figure 4D. First, the backbone of GSTE7, as predicted by AF, is in agreement with the experimental data, with an associated global deviation between both structures of 0.63 Å. The main AF/XRD structural differences are observed for residues Met53 (Met1 in GSTE7), Pro54 (Pro2), Gln191 (Gln124), Thr192 (Thr125), Ser315 (Ser222), and Asn316 (Asn223), which are all located in loops in the N- or C-terminals. We also compared local deviations computed using Cartesian coordinates (Figure 4D, top panel) with the same local deviation computed using internal coordinates, the coarse-grained angles (θ,γ) of the backbone [40]. These local probes are more suitable to analyze local deviations since they are not dependent on the algorithm used to perform the structural alignment between the structures. As shown in Figure 4D (middle panel), we found that structural differences observed using Cartesian coordinates for $\beta_{2}$ sheet and $\alpha_{2}$ helix are significantly reduced, which means that the local conformation of the backbone is in fact very similar. On the other hand, large local deviations observed for $\alpha_{4}$ and $\alpha_{5}$ helices from Cartesian coordinates are also very large, which confirms significant differences between AF and XRD structures of GSTE7 in this region. Last but not least, we performed the same study for the side-chain local conformations predicted by AF compared to XRD (Figure 4D, bottom panel). We used as internal coordinates $\chi_{1}$ dihedral angles as probes to characterize the local conformation of the side chains. Overall, there are many more differences in the side chain conformations compared to the backbone, which means that AF fails to predict the correct structure in specific locations. This is particularly true at the dimerization interface of GSTs. Moreover, as already observed from local deviation analysis of the backbone, the region between the $\alpha_{4}$ and $\alpha_{5}$ helices shows large deviation. Surprisingly, Ser168 (Ser104 in GSTE7) located in the “Wafer” motif has a $\chi_{1}$ distance of ≈1, which means that the local conformation is in opposite direction between the AF and XRD structures, equivalent to a cis vs. trans local conformation of this amino acid side chain in the experimental structure. By looking deeply into the XRD structure of GSTE7 (*4png*), there is a water molecule at the interface between the two monomers close to Ser104 that could play a role and also explain the orientation of the side chain. In addition, the temperature as well as the periodicity of the crystal are experimental conditions that may induce such differences. The relaxation of side chains predicted by AF using a classical force-field, for example, is therefore necessary if we are to delve deeper into the comparison between AI-predicted and experimental structures.

### 3.3. Characterization of the Dimerization Interface and of the GSH Binding Sites of GST Structures from Drosophila melanogaster

The two main structural features of GST related to their biological functions are the dimerization interface (DI) and the GSH binding site (GS). From the AF-predicted structures, we extracted residues which belong to the DI and to the GS (see Section 2.4) in order to characterize similarities/differences in the complete GSTome from *D. mel*. Figure 5 presents the study of the DI and, as shown in Figure 5A for all GSTs, residues in the DI are located in both domain I and II, essentially in the α helices. GST class with the largest group of residues in the DI is Sigma, with $N_{DI}>$ 40, whereas GST class with the smallest group of residues in the DI is Omega, with $N_{DI}<$ 20. For classes Delta, Epsilon, Theta, and Zeta, the number of residues in the DI is similar, between 25 and 35 residues. The total charge of the DI is a well-conserved property in the *D. mel* GSTome. Overall, the DI of GST structures is mostly positively charged and comprises between +2 and +6 for five of the six GST classes. The exception concerns GST class Omega, which presents mostly negatively charged DI (see Appendix A). Then, we identified MSA numbers presenting a singular behavior in *D. mel* GSTome. Particularly, we focus on MSA numbers 126, 127, 131, 158, and 162, which were found to belong to the DI of all 36 GST structures. These residues are located in the $\alpha_{3}$ and $\alpha_{4}$ helices and their nature in the different sequences of the GSTome is conserved or not, as depicted in Figure 5B. For instance, residues with MSA numbers 126, 131, and 158 are mainly hydrophobic residues interacting with each other, with conservation scores between 80 and 100%. Residues at MSA number 127 are mainly hydrophobic, with a majority of Tryptophan (in 55% of the GSTome), whereas residues at MSA number 162 are mainly hydrophilic residues (polar neutral), with a large majority of Glutamine (in 58% of the GSTome). As shown in Figure 5, the fact that these five residues are present in the DI of all the GSTs of *D. mel* with similar type of interactions means that there is a particularly high conservation of the DI for this organism during evolution; a modification of these amino acid by mutations, for example, might disrupt the homodimeric structure. In addition, 19 residues were found to belong to the DI of the large majority of the GSTome, with at least 30 of the 36 sequences involved. It confirms the high structural conservation of this feature, which is essential for dimeric GST to perform their catalytic activities. It corresponds to MSA numbers 108, 109, 124, 126, 127, 128, 130, 131, 135, 138, 154, 155, 158, 162, 165, 166, 168, 169, and 219. In addition, GST class Omega is characterized by less residues in the DI, specifically at the extremity of $\alpha_{4}$ helix and inside the $\alpha_{5}$ helix (Figure 5A). This is a consequence of the large separation observed between the two monomeric subunits, already mentioned in Section 3.2. On the other hand, GST class Zeta is characterized by a larger number of residues in the DI for these specific MSA numbers, which impacts the stability of the dimeric structures in this class.

To further investigate this observation, we computed the energy (in kJ/mol) and the surface area (in Å^2^) of the DI of all 36 structures in the GSTome, by taking into account the physical/chemical nature of the interactions using a classical force field (see Section 2.4). As shown in Figure 5C, the surface areas of the DI within a class are almost identical, whereas energies show a much larger variability. This means that the nature of the contacts between the residues at the interface is more relevant to characterize the DI than just their count. For instance, GSTS1 is the protein that presents both the largest number of residues in the DI and the largest surface area. However, GSTS1 is characterized by an energy of the DI very close to the median value of the GSTome. Furthermore, class Theta, which comprises enzymes that show surface areas of the DI very close to the median value of the GSTome, is one of the most stable classes. Finally, the most intriguing class is class Zeta. GSTZ1 and GSTZ2 show very similar sequences (sequence identity larger than 70%), very similar 3D folds (RMSD smaller than 2 Å), and very similar surface areas to those of the DI (≈2100 Å^2^). However, they are characterized by energies of the DI extremely different. GSTZ1 presents an energy of the DI around −1750 kJ/mol, whereas GSTZ2 presents an energy almost two times larger, i.e., around −3230 kJ/mol. To gain insights into this particular behavior of GSTZ2, we looked at the 3D atomic structure. As shown in Figure 5D, the two monomeric subunits are very close to each other, in contrast with class Omega. In addition, the DI does contain several Histidine and Glutamic acid, which involves strong electrostatic interactions between the monomeric subunits of GSTZ2. Such interactions have a major impact on the dynamics and flexibility of the protein and therefore in the biological function. For example, it is very unlikely that one would observe the breathing modes of the binding sites of GST for this class, as observed previously for GSTD1 and GSTD10 [28]. By comparing the DI residues for GSTZ1 and GSTZ2, differences are identified at MSA numbers 127–128, which have a Cys–Asp motif for GSTZ1 and an Ile–Glu motif for GSTZ2; MSA numbers 145–155 have a Pro–Val motif for GSTZ1 and a Val-His motif for GSTZ2; MSA numbers 166 and 179 have a Leucine and a Serine for GSTZ1 that are both substituted by Isoleucines in GSTZ2; MSA number 186 has an Isoleucine for GSTZ1 that is substituted by a Valine; MSA number 215 has a Glycine in GSTZ1 that is substituted by an Arginine in GSTZ2.

Next, a similar analysis was performed to characterize the GSH binding site (GS) of the complete *D. mel* GSTome (Figure 6). Structures were predicted using AlphaFold and AlphaFill, as described in Section 2.2 and summarized in the Appendix A. First, we found that residues in the GS are mostly located in domain I (Figure 6A), except for MSA numbers 170–180, located in the central $\alpha_{4}$ helix of GST structures. The total charge in the GS is also a well-conserved property in the *D. mel* GSTome. Overall, the GS total charge is slightly positive, between +1 and +2 for 86% of the GSTome, except for five GST structures of class Epsilon, i.e., GSTE1 and GSTE9–E12, which present a total charge of +3 or +4 (see Appendix A). Compared to the dimerization interface, there is no MSA number in the GSTome that shows a 100% identification score for the GSH binding sites. It comes from the fact that the N-terminal of GSTT3 structure is in the G site. It might be an artifact of AlphaFold, since, for this particular GST, the pLDDT is <30% and no GSH cofactors were found below sequence identity of 25% (see Section 2.2 and Appendix A). Nevertheless, four residues belonging to the GS of 35 out of 36 GST sequences were identified at MSA numbers 111, 112, 113, and 129, which are located in the loop between $\alpha_{2}$ helix and $\beta_{2}$ sheet in the thioredoxin fold, whereas MSA number 129 is located at the very beginning of the $\alpha_{3}$ helix. Residues at MSA numbers 111, 112, and 113 form a TVP sequence motif, which is highly conserved in the *D. mel* GSTome, particularly Pro113, which is conserved in 36 sequences of the GSTome. Val112 can be replaced by other hydrophobic non-aromatic amino acids such as Isoleucine; this is the case in GST class Delta. Thr111 is less conserved than the other two residues and can be replaced by positively charged Lysine; this is the case for the GST in classes Omega, Theta, or Zeta. This motif has been demonstrated to be involved in the stabilization of GSH in the GS [8]. We also identified four more residues at MSA numbers 64, 66, 110, and 128 that appear in the GS of 30 out of 36 sequences in the *D. mel* GSTome; this makes them of strong interest in the present work. Among them, we found a residue at MSA number 64, which corresponds to the active Serine already mentioned in Section 3.1. This residue is present in the GS of all GST structures except GSTD2, GSTD3, GSTS1, and GSTT3. Finally, class Omega does not include residues located in the $\alpha_{4}$ helix, was already the case for the DI, whereas class Zeta includes several residues located in this secondary structure. As explained above for energies of the DI, the fact that there are several contacts does not always mean that there are very strong interactions between residues and ligands. However, it is a good indicator of pocket volume accessible for the ligand to bind. Therefore, GSTs in class Zeta present binding pockets that appear to be more closed when compared to other classes.

Last but not least, we studied communication pathways between the two G sites of GST homodimeric structures. Basically, we identified residues involved in all possible existing pathways between the two GSH binding sites. Pathways were defined as a group of interacting residues that were visited to go from one GSH-bound site to the other APO (empty) site, as shown in Figure 6D. In this context, it has been shown experimentally that the presence of a GSH ligand in the one binding site induces modifications in the other monomeric subunit, corresponding to allosteric effects in GST [28]. From this analysis, all of the residues previously identified to belong to the GS were systematically involved in communication pathways. Moreover, several residues located in the middle of $\alpha_{4}$ helix were also identified. When compared to Figure 5A, this property is associated with the central motif in the dimerization interface. Surprisingly, none of the pathways identified residues located in $\alpha_{3}$ helix of the DI, which might be due to the presence of a GSH ligand in the pocket. Second, some GST sequences show residues in $\alpha_{5}$ and $\alpha_{6}$ helices but they are not highly conserved in the GSTome. For example, MSA number 243 in $\alpha_{6}$ helix is visited in GST structures of classes Omega, Sigma, and Theta only. The most-visited residue in the GSTome is the residue at MSA number 165, which is located in the middle of the $\alpha_{4}$ helix. This residue has been identified in the GS for the complete GSTome and its nature is class-dependent. It is mostly a Tyrosine for GST in class Delta, an Histidine for GST in class Epsilon, and a Glutamic or Aspartic acid in GST classes Omega, Theta, and Zeta. Figure 6D shows the structure of GSTZ2 with residues identified in the GS and an example of pathway passing through Glu165. It illustrates how the presence of a GSH ligand in the G site of one subunit involves a relatively short pathway to reach the other G site. Moreover, we saw that Glu165 (Glu114 in GSTZ2) in a given monomer is in contact with Glu128 (Glu81 in GSTZ2) of the other monomer. This is a very common interaction that has been identified in communication pathways of GST from the present work. Finally, by looking at the number of residues involved in communication pathways for each GST sequence, we observed that the average length is similar in the *D. mel* GSTome (≈30 residues), with GSTD3 that shows the shortest average pathway (20 residues) and GST in classes Sigma and Zeta that show the longest average pathways (≈40 residues).

To summarize, we identified an ensemble of key residues in *D. mel* GSTome, i.e., 10 in the DI and 8 in the GS that plays a crucial role. These residues, according to their MSA numbers, are the following: 108, 109, 124, 126, 127, 128, 130, 131, 135, 138, 154, 155, 158, 162, 165, 166, 168, 169 and 219 for the DI and 64, 66, 110, 111, 112, 113, 128, and 129 for the GS. Most of these residues have been highlighted in Figure 2A and their conservation plots are presented in the Appendix A. Among these residues, the-least conserved residue is at MSA number 168, with a maximum conservation score of 25% in the GSTome; this is very class dependent. In detail, it is a Methionine in GST class Delta; Serine in class Epsilon; Phenylalanine in class Omega; Histidine in class Theta; Cystein in class Zeta. The second least conserved residue is found at MSA number 165 (Figure 6B), which is located at the center of GST structures and close to the GS of both monomeric subunits. Its maximum conservation score is of 27% in the GSTome and this is also very class dependent, as described above. In addition, this specific location is part of “Clasp” and “Wafer” motifs in GST classes Delta and Epsilon and we also identified its crucial role in GST structures of the other classes. Another case can be highlighted, MSA number 130, which corresponds to positively charged amino acids in GST classes Delta and Epsilon; and hydrophobic residues in the other classes. Finally, MSA numbers 113 and 128 show much higher conservation, with Pro113 which belongs to the GS; these have perfect conservation in the GSTome, and have negatively charged Asp128 or Glu128. From our analysis, it is the only residue that is in both the DI and the GS of GST structures. The overall high conservation of these MSA number residues emphasizes their important role in GST from *D. mel*.

### 3.4. Analysis of the Structural Dynamics of GST from Drosophila melanogaster

From X-ray crystallography experiments, thermal B-factors, also called Debye–Waller factors or temperature factors, can be measured. They describe the attenuation of X-ray scattering caused by thermal motion. From AF-predicted structures, this information is not yet available, but physical models can be used to estimate structural fluctuations from atomic coordinates. One of them is normal mode (NM), which is a technique that can be used to describe flexible states accessible to a protein around its equilibrium position, in the harmonic approximation of small atomic displacements. In the present work, we applied an NM analysis (see Section 2.5) to predict thermal B-factors of the 36 AF-predicted structures from *D. mel*. First, we compared theoretical B-factors with experimental data (see Appendix A). For GST structures listed in Table 1, the comparison between the measured and predicted XRD thermal B-factors was very satisfying, with an average correlation of $\bar{\rho }=0.71$. The best prediction was obtained for GSTD2 (*5f0g*) with ρ=0.82; the worst prediction was obtained for GSTS1 (*1m0u*), with ρ=0.68 between the theoretical and experimental data. This particular behavior of GSTS1 is due to very large fluctuations measured experimentally in the region around MSA number 190, which corresponds to an external loop located between the $\alpha_{4}$ and $\alpha_{5}$ helices in domain II (see Appendix A). These fluctuations are not reproducible using NM calculations and are probably due to anharmonic oscillations in the structure.

Figure 7A presents predicted thermal B-factors of AF-predicted structures for the complete *D. mel* GSTome, as a function of MSA number. The most flexible regions are located in the N- and C-terminals, with B-factors values above 25 Å^2^, as envisioned from structural properties shown in Figure 2. On the one hand, residues which belong to the DI or to the GS show low flexibility for the complete GSTome (≈10 Å^2^) due to the fact most of these residues are located in α helices. On the other hand, some MSA numbers are associated with highly flexible region. For instance, MSA number 100 shows B-factors of ≈20 Å^2^ in all GST classes, except in class Omega. Similarly, MSA numbers 142–143 are highly flexible in class Delta and Epsilon but not much in the other classes. This result is not directly related to a difference of secondary structure as they are located in the linker loop between domains I and II. In terms of conservation, in GST class Delta, these MSA numbers correspond to a motif Lys–Asp and the polar amino acid is mostly a Serine in class Epsilon. However, it is less conserved in the other classes. This loop, which is located at the outside of the structure, is therefore more exposed to the solvent environment in GST structures of classes Delta and Epsilon and shows higher flexibility than the rest of the structure. This is also the case for the loop between the $\alpha_{4}$ and $\alpha_{5}$ helices, at MSA numbers 185–195. Furthermore, we computed the standard deviation of thermal B-factors in order to identify the MSA numbers that present the largest variability in flexibility/rigidity in the GSTome. Figure 7B represents both standard deviation and average thermal B-factors as a function of MSA number. Hereafter, we mainly focus on MSA number characterized with a few gaps in the MSA of the GSTome. First, structural fluctuations of the GSTome profile roughly vary between 10 and 20 Å^2^, apart from disordered N- and C-terminal parts. There are five mainly rigid regions (B<10 Å^2^), which correspond to the $\alpha_{1,2,3,4,5,6}$ helices plus the $\beta_{1}$ sheet, and which are associated with a low variability in the GSTome. The residues which belong to the DI and to the GS are located in these specific rigid regions. In addition, the $\alpha_{7}$ and $\alpha_{8}$ helices, located in domain II, show intermediate flexibility with thermal B-factors around 15 Å^2^, but are associated with a much larger variability compared to the α helices of the thioredoxin domain. In domain II, the largest variability is observed at MSA number 256. This MSA number has gaps in the Delta and Zeta classes. For the Omega class, it is located in the loop between the $\alpha_{6}$ and $\alpha_{7}$ helices, whereas in Epsilon and Theta classes, it is part of the $\alpha_{6}$ helix. It induces a larger flexibility in GST class Omega specifically.

In total, we identified six MSA numbers that are associated with both large variability and low number of gaps. This concerns numbers 142, 143, 193, 197, 198, 224, 263, and 281. MSA number 224, which is located at the extremity of the $\alpha_{5}$ helix, corresponds to a Glycine amino acid in 50% of the sequences, particularly in GST classes Delta and Omega. GSTs, which present residues other than Gly224, shows a lower flexibility in this region. MSA number 263, which is located between the $\alpha_{6}$ and $\alpha_{7}$ helices, is characterized by high flexibility in the Epsilon and Omega classes. In addition, it is not directly correlated with: (i) the nature of the amino acid, since, at this position, the maximum conservation score in the GSTome is 20%; (ii) the local conformation, since it is a loop for all GST structures in the GSTome. Within a class (Figure 7C), the flexibility profiles of the GST are pretty well conserved and GST classes showing the largest variability in thermal B-factors profiles are the Epsilon, Omega, and Theta classes. The GST Epsilon class presents a large variability at MSA numbers 310–314, which are associated with the extra β-sheet mentioned in Section 3.2 (Figure 3C). The GST Omega class at MSA number 200, located in the $\alpha_{5}$ helix, presents flexibility for GSTO1 and is characterized by an Aspartic acid, whereas for the other GST of this class, this residue is a Leucine or an Isoleucine. The GST Theta class presents a large variability in flexibility in the loop between $\alpha_{6}$ and $\alpha_{7}$. It is not relevant to discuss the variability in rigidity/flexibility in GST classes Sigma and Zeta since they only contain 1 and 2 structures, respectively.

Finally, we quantified the variability in flexibility/rigidity in the GSTome from *D. mel* by computing the Pearson correlation between thermal B-factors profiles of each GST individually (Figure 7A) with the average profile of the complete GSTome (Figure 7B). As shown in Figure 7D, GST classes Delta, Epsilon, and Theta are characterized by correlations ρ>0.9. However, GSTE10 and GSTT3 show singular behaviors, with correlation much smaller than the other GST structures of the same class. As shown in Figure 3, the N-terminal of GSTT3 is very long and not well predicted by AF. In addition, it shows a lower flexibility in this region compared with the average profile. The same analysis holds for the very long C-terminal helix of GSTE10. GST classes Zeta and Sigma show correlation values $\bar{\rho }=0.87$ and $\bar{\rho }=0.82$, respectively. Finally, GST class Omega shows correlation $\bar{\rho }=0.78$, which is the smallest among the GSTome. This means that the GST from the Omega class are characterized by a flexibility profile slightly different compared to the other classes; this correlates with the fact that they present a 3D fold that is different from the other GSTs from the *D. mel* (see Section 3.2).

## 4. Conclusions and Perspectives

In the present work, we used AlphaFold to predict the 36 structures which belong to the GSTome of *Drosophila melanogaster*. From a complete structural analysis, we observed that the global folds in the GSTome are very similar within a class, except some additional N- and C-terminals, and are characterized by identical patterns in terms of secondary structures. With the exception of GSTD3, which does not exhibit a β-sheet in its N-terminal, all of the GSTs comprise a βαβαββα thioredoxin-like fold (domain I). However, domain II is characterized by more structural differences depending on GST classes. It mainly comprises α helices in the GSTome, with the exception of GSTE1–GSTE10, which present two short β sheets between the $\alpha_{4}$ and $\alpha_{5}$ helices in its C-terminal. Depending on the sequence, the α helix, denoted as $\alpha_{7}$, can either be considered as one or two separated helices, as was observed for GSTE7 and GSTS1, respectively. Between GST classes, significant structural differences were identified; these were especially located in the α helical domain of the GSTs. Here, the thioredoxin-like fold was locally conserved even for the GSTs which exhibited differences in their 3D folds. Compared to the experiment results, GST structures predicted by AlphaFold are in very good agreement with the XRD structures, as shown for GSTE7. Only slight differences were observed in backbone conformation, in the region between the $\alpha_{4}$ and $\alpha_{5}$ helices. However, larger differences were observed in the side-chain orientation between the predictions and the experiments, which can limit the benefit of AlphaFold; this is particularly in cases where one is attempting to predict the dimerization interface or the GSH binding site in the GST in detail.

Then, from the AF-predicted structures, we characterized the two main structural features of the GSTs, i.e., the dimerization interface and the GSH binding site, by identifying key residues. For the DI, we identified 19 key residues in the MSA and we found that the $\alpha_{3}$ helix in domain I and the $\alpha_{4}$ helix in domain II are strongly involved in the interactions between monomeric subunits for the complete GSTome. However, the nature of amino acids in these specific regions is strongly dependent on GST classes. This characteristic has already been observed experimentally for GST classes Delta and Epsilon, with the well-known “Clasp” and “Wafer” motifs. From the present work, this characteristic can be extended to the other GST classes Omega, Sigma, Theta, and Zeta. We also identified that the GST Zeta class presents a unique G site and dimerization interface compared to the others, with interaction strength between the two monomeric subunits of GSTZ2 being much larger than those of any other GST. In a similar manner, we identified residues that belong to the GS using Gluthatione cofactors predicted by AlphaFill. We identified eight key residues in the MSA, with residues essentially located in domain I and associated with very high conservation scores in the GSTome. Particularly, residue at MSA number 111 was systematically found to belong to the GS in the GSTome and is highly conserved, with a Threonine amino acid in the GST Delta and Epsilon classes and a Lysine amino acid in the GST Omega, Sigma, Theta, and Zeta classes. From a communication pathway analysis between the GSs, we found that the central residue at MSA number 165 establishes a connection between the two binding sites, when one GSH ligand is bound to one site and the other is APO. In addition, the nature of this residue is class-specific, with Tyrosine, Histidine, and Glutamic acid residues in the GST Delta and Epsilon classes and all the others, respectively. We can emphasize that drastic differences in the nature of amino acid along the sequence, from hydrophobic to charged amino acids in such a specific location in the sequence play a crucial role in the catalytic properties of the GSTs and can be associated with different biological functions.

Finally, we predicted the dynamics and flexibility profiles of the *D. mel* GSTome. We found that regions of both low flexibility and low variability are associated with the α helices, which contain residues of the DI and the GS. We showed that regions of both high flexibility and variability are either N- or C-terminal regions and we found loops that are located towards the outside of GST structures which are exposed to the solvent environment. We showed that these are regions with largest fluctuations between NM predictions and thermal B-factors obtained from X-ray crystallography. MSA numbers of medium flexibility and large variability are associated with thermal B-factor profiles that are very specific to GST classes. It is either associated with a difference in the secondary structure or to a difference in the local conformation, as repeatedly observed in the linker loop between GST classes Epsilon and Omega. *In detail*, all these data extracted from the GSTome from *Drosophila melanogaster* helped us to better understand the relationship between sequences, structures, and conservation during evolution. The data will also guide us to design synthetic GSTs with new/optimal catalytic properties for detoxification applications, and particularly determine how spontaneous mutations/insertions/additions in the sequences modify the stability, selectivity, and catalytic efficiency of the GST enzymes.

## Figures and Tables

**Figure 1 biomolecules-14-00759-f001:**
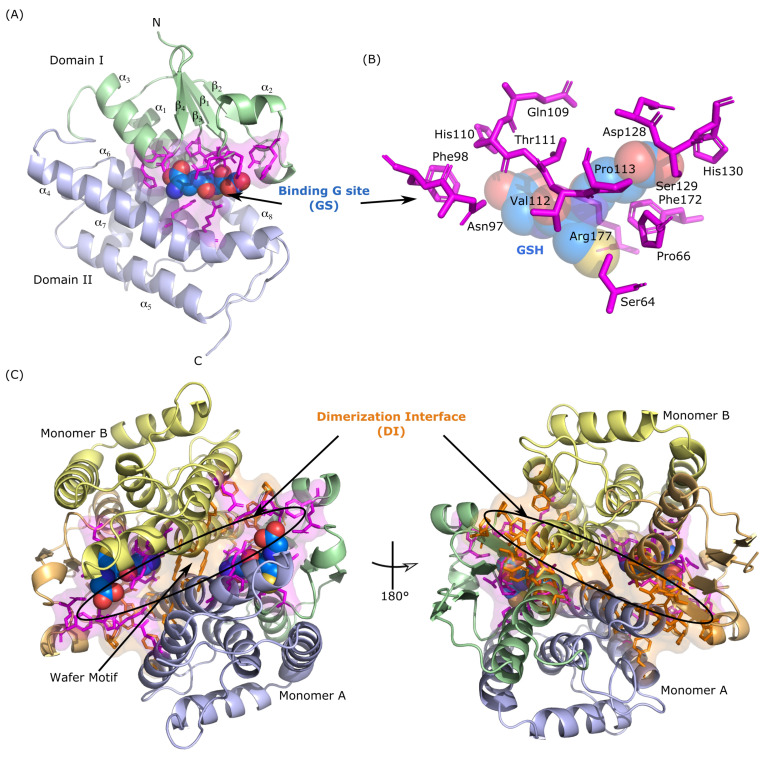
(**A**) Structure of GSTE7 monomeric subunit. The thioredoxin-like N-terminal domain (I) is shown in green and the α helical C-terminal domain (II) is shown in blue. GSH ligand is shown with blue spheres and residues belonging to the binding site G are represented with magenta sticks. (**B**) Structure of GSH binding site of GSTE7. The color code is the same as in panel (**A**), with labels indicating residue name and MSA number. (**C**) Visual representation of the homodimeric structure of GSTE7. Monomer A is represented as performed in panel (**A**) and monomer B is represented in yellow/orange. Residues belonging to the dimerization interface are represented with orange sticks. The “Wafer” lock and key motif is also indicated.

**Figure 2 biomolecules-14-00759-f002:**
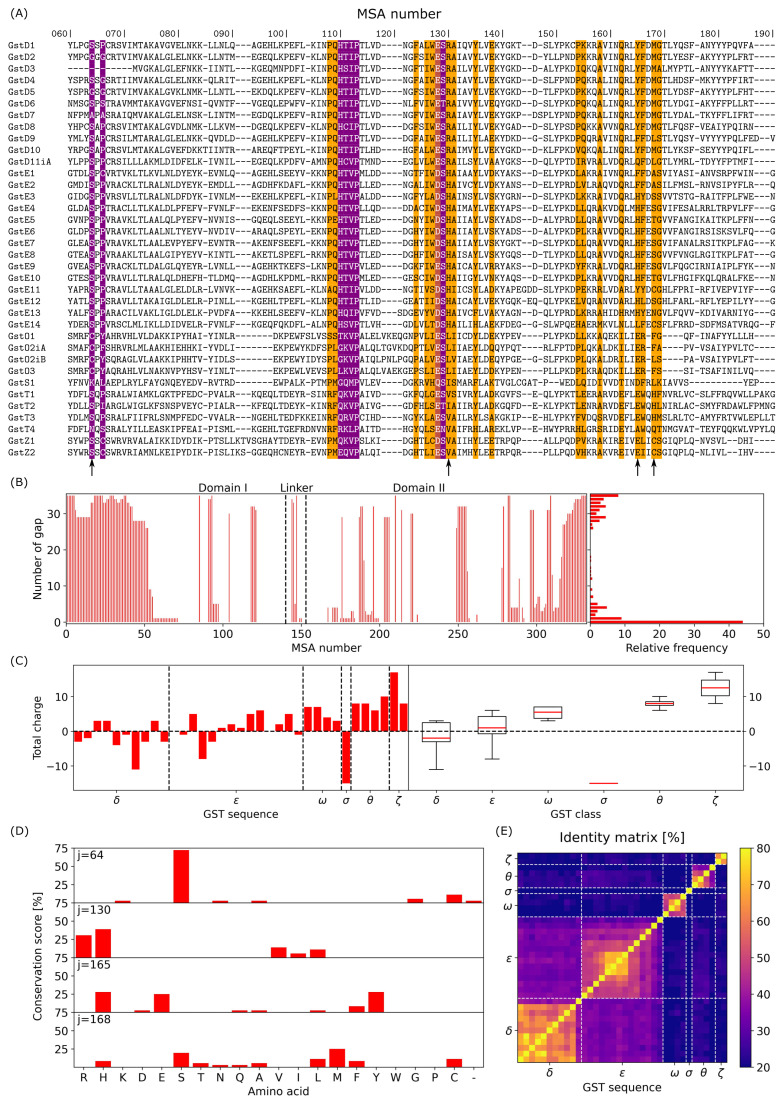
(**A**) Sample of multiple sequence alignment of *D. mel* GSTome from MSA numbers 60 to 190. Text highlighted in magenta and orange corresponds to residues which belong to the GSH binding sites and the dimerization interface, respectively (see Section 3.3). (**B**) Number of gaps as a function of MSA number and histogram of its relative frequency. (**C**) Total charge of the 36 sequences of GST in the *D. mel* GSTome (left panel) and as a function of GST class (right panel). (**D**) Conservation score as a function of the 20 natural amino acids for various MSA numbers 64, 130, 165, and 168. (**E**) Sequence identity matrix for the complete *D. mel* GSTome.

**Figure 3 biomolecules-14-00759-f003:**
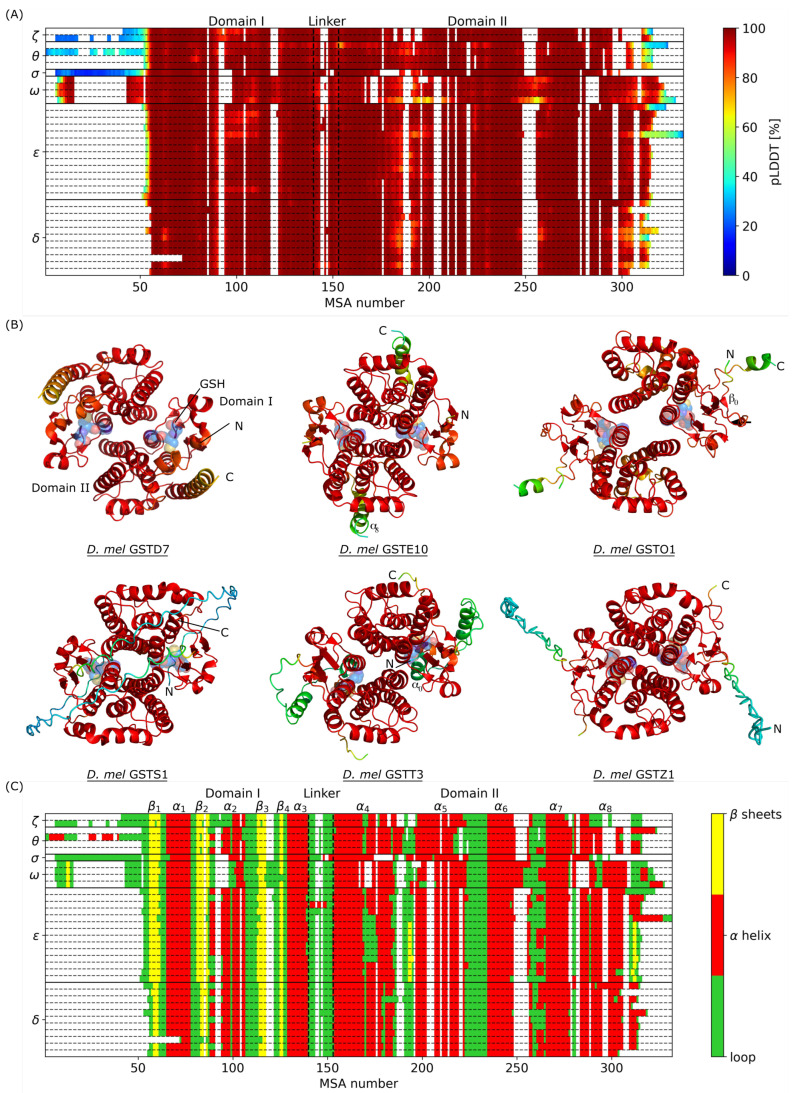
(**A**) Prediction score (pLDDT) as a function of MSA number. (**B**) Visual representation of GST structures from *D. mel* predicted by AlphaFold. One structure per class is shown, the others are shown in the Appendix A. The color code corresponds to pLDDT values given in panel (**A**). GSH cofactors are represented with blue transparent spheres. (**C**) Secondary structure prediction as a function of MSA number. Loops are colored in green, α helices in red, and β sheets in yellow.

**Figure 4 biomolecules-14-00759-f004:**
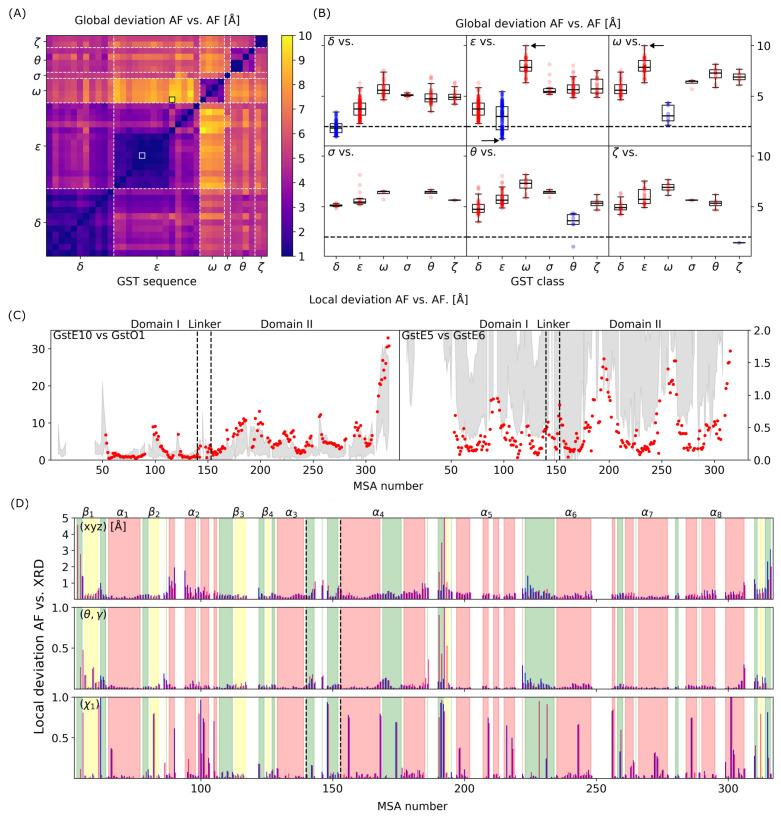
(**A**) Global deviation (RMSD matrix, in Å) between pairs of AlphaFold-predicted structures of *D. mel* GSTome. (**B**) Statistical analysis of global deviation (in Å) as a function of GST class. (**C**) Local deviation (in Å) as a function of MSA number for pairs GSTE10 vs. GSTO1 (left panel) and GSTE5 vs. GSTE6 (right panel). Gray area represents the standard deviation of local deviation computed for the complete GSTome. (**D**) Local deviation (in Å) as a function of MSA number between AlphaFold predicted and XRD measured structures for GSTE7. Secondary structures are highlighted using the same color code as in Figure 3C. Top, middle, and bottom panels indicate local deviations computed using Cartesian coordinates, internal coordinates (θ,γ) angles of the backbone and internal coordinates $\left(\chi_{1}\right)$ angles of the side chains, respectively.

**Figure 5 biomolecules-14-00759-f005:**
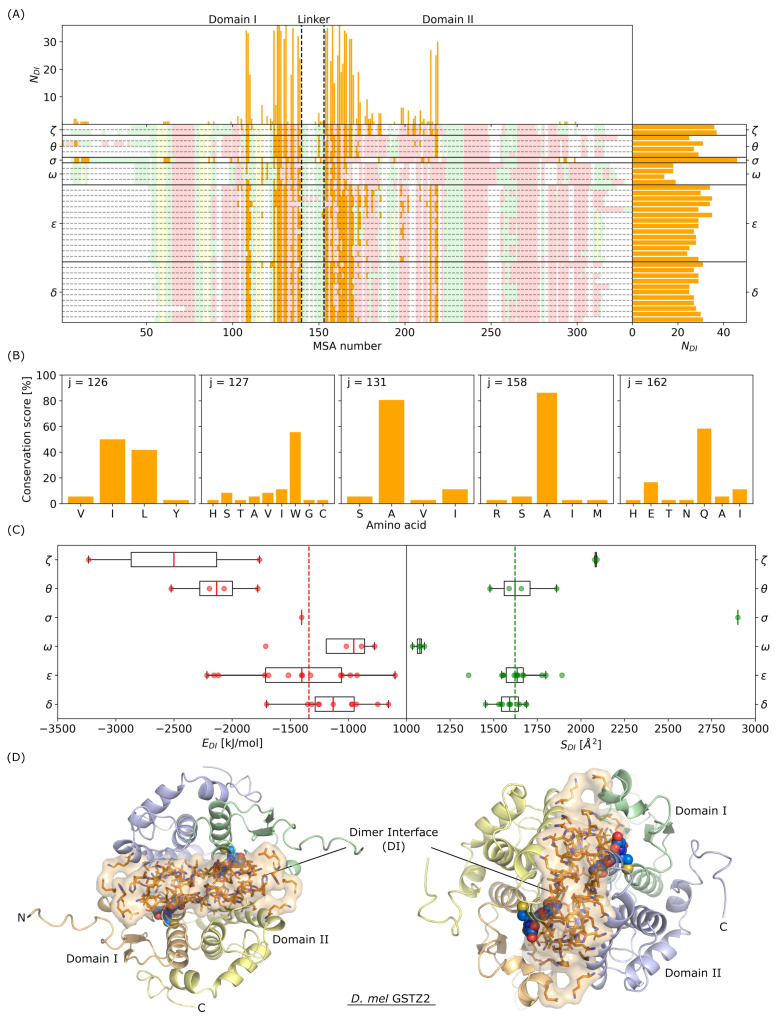
(**A**) Number of contacts $N_{DI}$ in the dimerization interface of GST as a function of MSA number. Secondary structures are indicated using the same color code as in Figure 3C. (**B**) Conservation score (in %) as a function of the amino acid nature for MSA numbers 126, 127, 131, 158, and 162. (**C**) Box plot of energies and surfaces of the dimerization interface for the complete GSTome of *D. mel*. Dashed lines indicate median values of the complete GSTome. (**D**) Visual representations of GSTZ2 structure predicted by AF. The color code is the same as in Figure 1. Residues identified in the dimerization interface are shown in orange.

**Figure 6 biomolecules-14-00759-f006:**
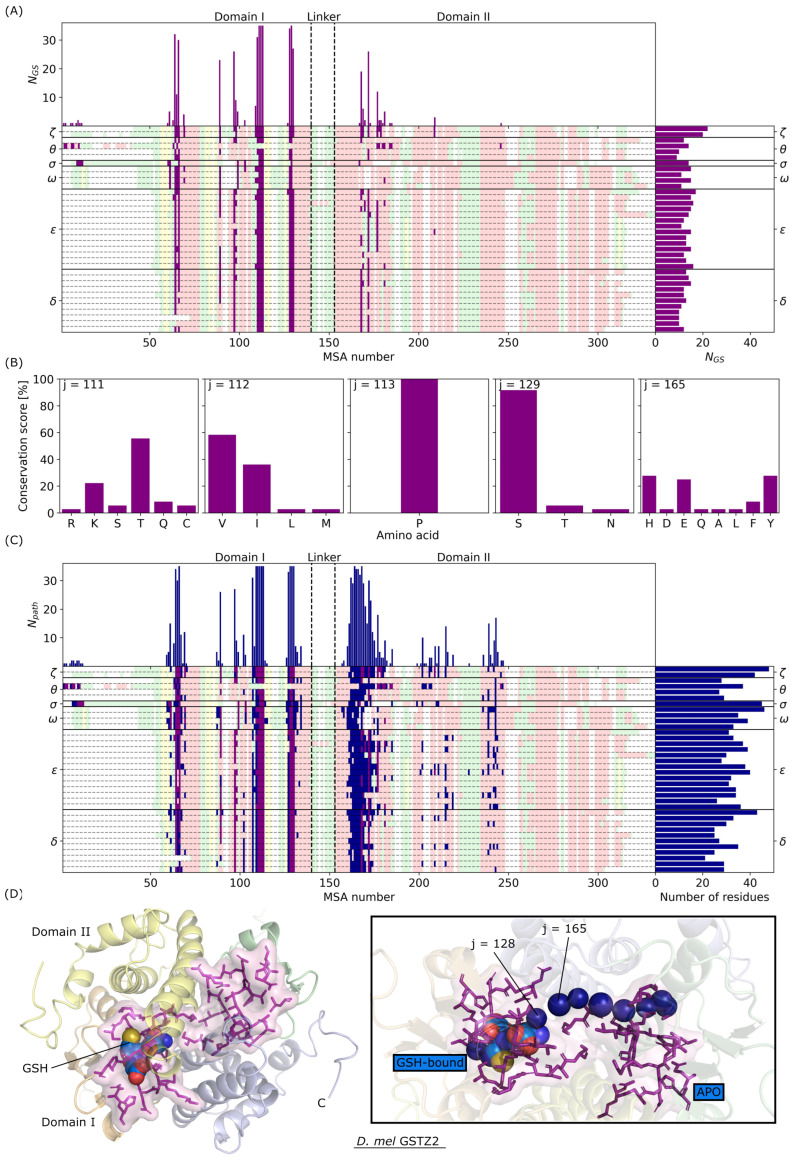
(**A**) Number of contacts $N_{GS}$ in the GSH binding site (GS) as a function of MSA number. Secondary structures are indicated using the same color code as in Figure 2C. (**B**) Conservation score (in %) as a function of the amino acid nature for MSA numbers 111, 112, 113, 129, and 165. (**C**) Number of residues found in the communication pathways $N_{path}$ between the two G sites of GST as a function of MSA number. (**D**) Visual representations of GSTZ2 structure predicted by AF. The color code is the same as in Figure 1. Residues identified in the GSs are shown in magenta.

**Figure 7 biomolecules-14-00759-f007:**
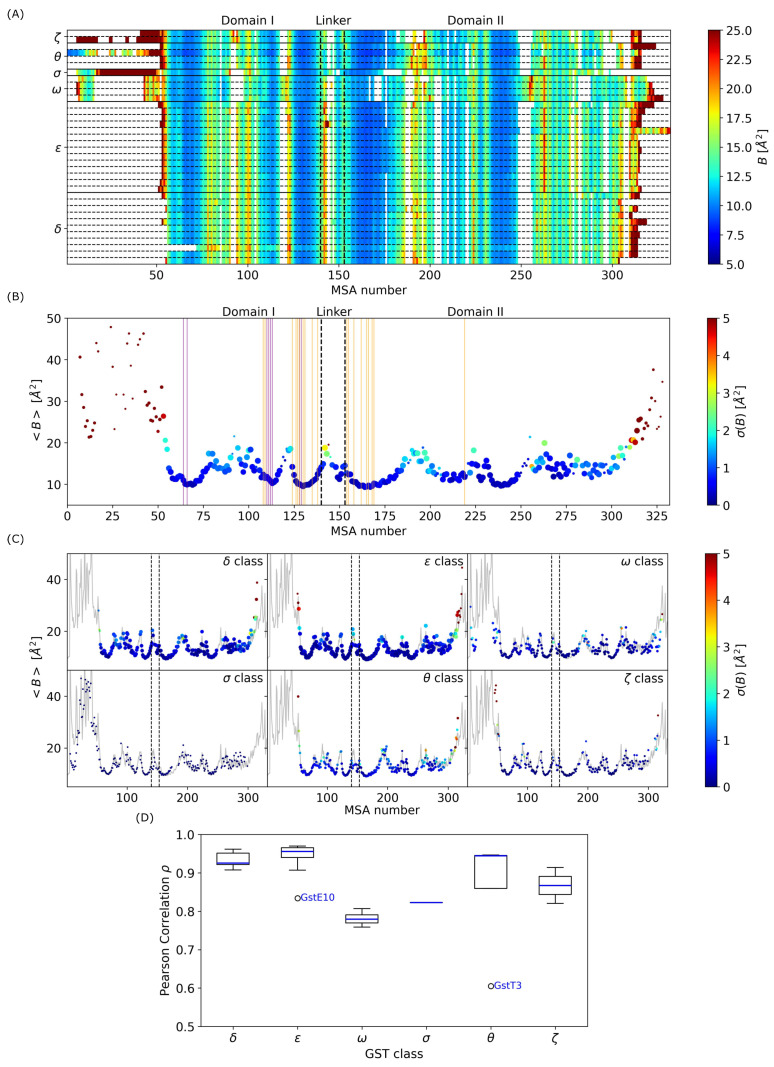
(**A**) Thermal B-factors (in Å^2^) of GST structures as a function of MSA number. (**B**) Average B-factors (in Å^2^) as a function of MSA number for the complete *D. mel* GSTome. Circles are colored according to the standard deviation of B-factors in the GSTome; their corresponding sizes indicate the occurrence of each amino acid in the sequence at a given MSA number. The size can vary from 1 to 36. Orange and purple lines indicate the position of residues belonging to the DI and to the GS, respectively. (**C**) Average B-factors (in Å^2^) as a function of MSA number for each class of the GSTome. Grey lines represent the average thermal B-factors, as shown in panel (**B**). (**D**) Pearson correlation ρ as a function of GST class between B-factors of a given sequence and the average B-factors of the complete *D. mel* GSTome shown in panel (**B**).

## Data Availability

The authors confirm that the data supporting the findings of this study are available within the article [and/or] its Appendix A.

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
