# Peer review of "Structural Analysis of the Drosophila melanogaster GSTome"

_biomolecules, 2024, doi:10.3390/biom14070759_

Round 1
Reviewer 1 Report
Comments and Suggestions for Authors
The authors have compiled the excellent piece of work. The article is insightful with comprehensive information. Few additional information might be included.
1. Is there any difference in the electrostatic charges between the 36 structures?
2. If any differences in hydrophobic, or electrostatic charge, does it impact the ligand binding.
3. Structural changes among the different classes of the GST can be included as Table for easy comparison.
Typo error in Page 8 , Line 8: ‘Root-Mean-Squared-Deviation od”.
Reviewer 2 Report
Comments and Suggestions for Authors
This paper is an extremely detailed examination of the D. melanogaster GSTome: 36 varieties of glutathione transferase (GST), in 6 classes. 3-D structures were generated using AlphaFold, with good validation from the available XRD structures, plus AlphaFill to place bound GSH molecules. Flexibility was calculated from normal mode analysis, and correlated with B-factors from crystallography. Properties analyzed included interactions between monomers (GSTs are homodimers), structure of the GSH binding site and of the channel between sites, flexibility. All were analyzed using 3-D structure in conjunction with multiple sequence alignment, both within and between classes.
The study is a good example of an in-depth analysis (making good use of the capabilities of AlphaFold/AlphaFill) of a large set of proteins, related by evolution and of similar overall construction, but with a variety of biological functions. The extensive structural details, plus the final summary, will help to guide design of synthetic GSTs for detoxification applications.
The figures are well-made and informative. There is a lot of information crammed into them, but that is inevitable with this type of paper.
There are a few minor problems with the text:
Section3.2: Structures 3ein, etc. are described as monomeric - they are actually dimeric, but there is only one monomer per asymmetric unit; the second one is a symmetry mate. Also, at line 15 residues 127 and 162 are described as mainly hydrophobic, but residue 162 appears mainly hydrophilic.
Fig. 6B caption gives the wrong sequence numbers.
Fig. 7B caption says "size of the dots indicates number of gap induced by the MSA" - what does this mean?
Comments on the Quality of English LanguageThe English is very good. There are a few places where singular and plural do not agree between noun and verb, and typos such as "the the" where "of the" is meant, and misspelling of "pollutants".
